# The Influence of Metabolic Factors in Patients with Chronic Viral Hepatitis C Who Received Oral Antiviral Treatment

**DOI:** 10.3390/metabo13040571

**Published:** 2023-04-17

**Authors:** Oana Irina Gavril, Radu Sebastian Gavril, Florin Mitu, Otilia Gavrilescu, Iolanda Valentina Popa, Diana Tatarciuc, Andrei Drugescu, Andrei Catalin Oprescu, Andreea Gherasim, Laura Mihalache, Irina Mihaela Esanu

**Affiliations:** 1Department of Medical Specialties (I), Faculty of Medicine, “Grigore T. Popa” University of Medicine and Pharmacy, 700115 Iași, Romania; 2Academy of Medical Sciences of Romania, Bucuresti 030171, Romania; 3Department of Medical Specialties (II), Faculty of Medicine, “Grigore T. Popa” University of Medicine and Pharmacy, 700115 Iași, Romania; 4Morpho-Functional Department, Faculty of Medicine, “Grigore T. Popa” University of Medicine and Pharmacy, 700115 Iași, Romania

**Keywords:** metabolic syndrome, chronic hepatitis C virus, sustained viral response

## Abstract

Hepatic diseases pose a significant public health concern. Regardless of the severity of hepatic fibrosis, treatment is recommended for all chronic hepatitis C virus (HCV) subjects. However, fibrosis and steatosis assessment remains crucial for evaluating the prognosis, progression, and hepatic disease monitoring, particularly following the treatment with direct-acting antivirals (DAAs). The aim of our study was to evaluate the impact of metabolic factors and the extent of hepatic fibrosis and fat accumulation in chronic HCV infection subjects. Additionally, another objective was to investigate modifications regarding fibrosis and steatosis three months after a successful sustained viral response (SVR). A total of 100 patients with compensated cirrhosis and chronic hepatitis C (CHC) were included in our study. These patients received treatment with DAA and underwent Fibromax assessment before and three months post SVR. After DAA treatment, a significant decrease was observed in the degree of hepatic fibrosis and hepatic steatosis. This regression was evident three months following the achievement of SVR. Chronic viral hepatitis C may trigger risk factors for metabolic syndromes, such as obesity and type 2 diabetes mellitus. Conclusions: It is crucial to monitor metabolic factors and take timely measures to prevent or treat metabolic syndrome in patients with chronic viral hepatitis C.

## 1. Introduction

Global estimates indicate that 71 million people suffer from hepatopathies caused by chronic hepatitis C virus (HCV) infection. The prognosis and disease progression have significantly improved due to the availability of Direct-acting antiviral (DAA) therapy [1,2]. The global accessibility of DAA treatment and the simplification of therapeutic strategies have made it possible to eliminate HCV by improving screening methods, identifying infected individuals with active infection, and initiating timely treatment. Considering the effectiveness of DAA treatment and the favorable drug tolerance, elimination (without a vaccine eradication cannot be achieved) is possible, provided screening methods and treatment accessibility are improved. Oral antiviral treatment for hepatitis may have some side effects on other organs, but these are usually minor. However, the treatment may be beneficial for other organs, as it can reduce inflammation and oxidative stress. Regarding side effects related toother organs, antiviral medications may have a negative effect on the kidneys, cardiovascular system, and central nervous system. However, these side effects are rare and usually only occur in patients who already have serious comorbidities. It is important that patients are closely monitored during antiviral treatment so that any side effects can be identified and treated. Overall, however, oral antiviral treatment is considered safe and effective in treating hepatitis and has significant benefits for overall health [3].

The World Health Organization (WHO) estimates that elimination, as defined by them, can be achieved by up to 80% by 2030.Despite the significant breakthrough of DAA therapy in the treatment of HCV, with at least 95% of individuals achieving a sustained virologic response (SVR), the evolution remains uncertain due to the significant risk of hepatocellular carcinoma and the progression of hepatic disease. The severity of fibrosis and steatosis influences the post-SVR evolution and the possibility of hepatocellular carcinoma risk. HCV has been the leading cause of liver transplantation. However, it is currently being replaced by nonalcoholic fatty liver disease (NAFLD) and hepatocellular carcinoma caused by toxic hepatic disease [4,5].

Hepatic steatosis in persons infected with HCV may also be caused by other factors among patients who are not infected with HCV. For example: metabolic syndrome, increased body mass index (BMI), hypertriglyceridemia, chronic alcohol consumption, other infections, or exposure to certain medications [6,7]. Alcohol consumption enhances viral replication in liver diseases and accelerates disease progression. Even moderate amounts of alcohol can influence the degree of hepatic fibrosis and steatosis. Patients with liver disease are advised not to consume alcohol [8]. Unlike hepatic steatosis caused by genotype 3, that caused by metabolic syndrome or insulin resistance is associated with accelerated fibrosis progression, decreased response to treatment, and increased risk of hepatocellular carcinoma (HCC) [9]. In many subjects, both viral and non-viral factors contribute to the development of hepatic steatosis. Indeed, HCV replication relies on lipid metabolism for its replication cycle and leads to hepatic steatosis through multiple mechanisms, such as increased lipogenesis, altered mitochondrial lipid oxidation, and decreased activity of microsomal triglyceride transfer protein [10]. Hepatic steatosis itself can increase HCV replication, which has been shown to be present in alcohol-induced lipid metabolism alteration [11]. Hepatic steatosis represents a common histologic feature in chronic viral hepatitis C, but it can also be present independently, in association with obesity, increased alcohol consumption, type 2 diabetes mellitus (T2DM), and hyperlipidemia. These can contribute to the occurrence of hepatic steatosis in patients with HCV infection. In many studies, the link between hepatic steatosis and BMI has been highlighted, an association that is significant, especially in patients with HCV genotype 1 [12]. This has led to the idea that in patients with HCV viral infection, there is “metabolic fat” (especially in patients with HCV genotype 1) and “viral fat” (especially among patients with genotype 3). Epidemiological studies suggest a link between T2DM and HCV infection [13].

The oral antiviral treatment for patients with chronic viral C hepatitis is adapted according to current guidelines for this pathology (treatment for 8, 12, or 24 weeks depending on the severity of the disease). There are six genotypes and over 100 subtypes of HCV known. Among these, genotypes 1, 2, and 3 are found worldwide.

All subjects in our study presented genotype 1b, which predominates in Romania with a percentage of 99%.

One of the factors that determine the progression of HCV infection is alcohol consumption—it increases HCV replication, accelerates disease progression, and accelerates liver damage. Even moderate amounts of alcohol influence the degree of liver fibrosis. Although the American Gastroenterological Association recommends that individuals at high risk who do not have evidence of liver cirrhosis may be excluded from follow-up if the Fibroscan value is less than or equal to 9.5 kPa post-SVR, individuals at high risk diagnosed with pre-treatment liver cirrhosis or with other risk factors for chronic liver disease (obesity, T2DM, human immunodeficiency virus, co-infection with hepatitis B virus, excessive alcohol consumption) may be misclassified as not having advanced fibrosis in 6.6% of cases and still require monitoring [14]. Accurate interpretation of non-invasive assessments of fibrosis in post-SVR patients is essential and requires ongoing investigation.

## 2. Materials and Methods

We conducted a prospective study on 100 patients who had viral hepatitis C infection, including both newly diagnosed and previously known cases of liver disease. The study was conducted atthe Institute of Gastroenterology, “St. Spiridon” Hospital, Romania between 2018 (January) and 2020 (March). The subjects were investigated before and post SVR (after three months) based on imaging, biological (fasting blood tests), and clinical criteria such as upper gastrointestinal endoscopy andan abdominal ultrasound exam. The first evaluation occurred before antiviral treatment, while the second visit occurred three months post-SVR (T3). The DAA therapy consisted of either ombitasvir/paritaprevir/ritonavir + dasabuvir or ledipasvir + sofosbuvir. We considered chronic alcohol consumption subjects who consumed more than 100 g of alcohol per week.

The study enrolled patients who met the following inclusion criteria, in accordance with national and international guidelines: 18 years or older, with a positive RNA-HCV test, and who provided informed consent. Exclusion criteria included patients with undetectable HCV RNA, those who were not recommended for antiviral treatment because of their comorbidities, those with advanced forms of liver disease (indicated by clinical, biological, and imaging exams) such as hepatic encephalopathy, variceal gastrointestinal bleeding, ascites or jaundice, and those with hepatocellular carcinoma or other malignancies. All patients provided written informed consent before enrollment, and the research was conducted in agreement with the principles of the Declaration of Helsinki and received approval from the “Grigore T. Popa” University Ethics Committee, Iasi, Romania.

All patients who participated in the study were required to have 1b genotype with a complete liver panel to assess liver disease, detectable viremia, comorbidities evaluation, and approval from a specialist doctor for any other conditions. In addition, the degree of liver fibrosis was evaluated using a noninvasive test called Fibromax (the tests were sent to a certified laboratory—BioPredictive), both before treatment initiation and three months after achieving SVR. To ensure consistency in diagnosis, all enrolled patients underwent testing using the same method.

Fibromax is a blood test that requires a blood sample, minimum 2 mL, taken in the morning. Immunonephelometry and photometry were the techniques used to process the samples. Fibromax comprises five tests (noninvasive): FibroTest, which evaluates the liver fibrosis severity; ActiTest, which assesses the activity of necroinflammation; SteatoTest, which determines the hepatic steatosis degree; NashTest, which evaluatesifnonalcoholic steatohepatitis is present among metabolic syndrome subjects; and AshTest, which assesses the extent of liver damage in individuals who chronically consume alcohol [15,16,17,18].

Formulas correlated with date of birth, sex, height, and weight are used to calculate the results of the five noninvasive tests included in Fibromax: FibroTest, ActiTest, SteatoTest, NashTest, and AshTest. Blood tests including aspartate aminotransferase, alanine aminotransferase, gamma-glutamyltranspeptidase, total bilirubin, fasting serum glucose, haptoglobin, alpha 2 macroglobulin, apolipoprotein A1, triglycerides and cholesterol are assessed using this score. For the current study, only the results of FibroTest and SteatoTest were taken into account. The local laboratory used a specific calibration for delimiting the degrees of liver fibrosis and steatosis based on the FibroTest and SteatoTest scores. For instance, FibroTest score cut-offs for identifying the stages of liver fibrosis are F4 > 0.75, 0.5 < F3 ≤ 0.75, 0.25 < F2 ≤ 0.5, and F1 ≤ 0.25, while SteatoTest score cut-offs for delimiting the stages for liver steatosis are S3 > 0.75, 0.5 < S2 ≤ 0.75, 0.25 < S1 ≤ 0.5, and S0 ≤ 0.25.

The National Health Fund established the following criteria: patients with fibrosis levels F2, F3, and F4 were eligible for DAA initiation, while those with F0 and F1 fibrosis were not.

In order to conduct statistical analysis, version 18.0 of SPSS software was utilized. The data were analyzed descriptively using ANOVA. To compare frequency distributions within or between groups, the non-parametric Chi-Square and Kruskal–Wallis tests were employed. The normal range of values was assessed prior to applying tests for statistical significance, with continuous variables determined by the Skewness test. ANOVA test results provided information on mean value indicators (e.g., maximum and minimum values, median, mean, modulus) and dispersion indicators (e.g., standard deviation, standard error, coefficient of variation). To compare continuous variables across different groups, a significance threshold of 95% (*p* < 0.05) was used with Student’s *t*-test and paired-samples *t*-test. Multiple comparisons of normally distributed value series were performed using a post-hoc Bonferroni test following one-way ANOVA.

## 3. Results

The study population comprised 100 individuals diagnosed with chronic hepatitis C infection, with 65% of them being female. The age range of the patients varied from 35 to 77 years. The mean was close to the median of the group, which indicated a uniform distribution of values, and this was corroborated by the Skewness and Kurtosis tests. Therefore, statistical tests of significance for continuous variables were applicable. The distribution of the participants by age and gender did not indicate any significant differences (*p* = 0.089). The treatment was administered to 72 patients with ombitasvir/paritaprevir/ritonavir + dasabuvir, while the remaining 28 received ledipasvir + sofosbuvir treatment. The average value for fibrosis was 0.65 ± 0.18, ranging from 0.32 to 0.96, with a median value of 0.65, which suggested a homogenous distribution of the values and was supported by the Kurtosis and Skewness tests. As a result, tests for statistical significance were conducted for the continuous variables. Grade F4 fibrosis was observed in the majority of cases (43%) in the study. The range of steatosis values was from 0.11 to 0.89, with a mean value of 0.50 ± 0.18, which was in proximity to the group’s median value (0.49). Skewness and Kurtosis tests indicated a uniform distribution of values, which allowed for the use of statistical tests of significance for continuous variables. The majority of cases had grade S2 steatosis (37%) (Table 1).

After a period of three months after achieving SVR, a notable reduction was noted in the level of both hepatic steatosis and fibrosis (Table 2).

From the total study sample, 11% of patients were chronic alcohol consumers, 9.2% of women and 14.3% of men, with chronic alcohol consumption representing a slightly higher risk factor for males (RR = 1.35; 95% CI: 0.66–2.74; *p* = 0.448) (Figure 1):

Clinical parameter evolution in patients with fibrosis

Regarding the evolution of laboratory parameters at a minimum of 3 months after achieving SVR, the following aspects were highlighted (Table 3):In patients with mild fibrosis

Initially, the average BMI was significantly lower (*p* = 0.043), which remained so after SVR (*p* = 0.034);

The average level of triglycerides (*p* = 0.013) significantly increased;

The average level of gamma-glutamyl transferase (GGT) (*p* = 0.001), alanine transaminase (ALT) (*p* = 0.001), and aspartate aminotransferase (AST) (*p* = 0.001) significantly decreased (to normalization).

In patients with severe fibrosis

The mean levels of GGT (*p* = 0.001), AST (*p* = 0.001), and ALT (*p* = 0.001) decreased significantly

Evolution of clinical parameters in patients with steatosis

Regarding the evolution of laboratory markers post-treatment in patients with steatosis, the following aspects were highlighted (Table 4):Patients with mild steatosis:

Initially, the mean BMI was significantly lower (*p* = 0.001), registering a significant increase after SVR (*p* = 0.004);

The mean level of triglycerides (*p* = 0.001) significantly increased;

The mean level of GGT (*p* = 0.005), ALT (*p* = 0.001), and AST (*p* = 0.001) significantly decreased (until normalization).

Patients with severe steatosis:

The mean level of GGT (*p* = 0.002), ALT (*p* = 0.001), and AST (*p* = 0.001) significantly decreased.

## 4. Discussions

In our research, all the included subjects presented genotype 1b. These data are supported by other epidemiological studies that highlight the higher prevalence of genotype 1, followed by genotype 3 [19]. Studies show that a much higher prevalence of hepatic steatosis is encountered in HCV genotype 3 infections, compared to non-genotype 3 HCV (74% versus 48%) [20].

Our study aimed to evaluate the regression of fibrosis and steatosis severity using Fibromax (a noninvasive method), both before treatment and after achieving sustained virologic response (SVR) through DAA treatment. Previous studies have investigated this relationship using hepatic elastography and percutaneous liver biopsy, with the latter being the preferred method for evaluating the severity of steatosis and fibrosis [21,22]. Most patients with HCV who undergo DAA treatment achieve SVR [23]. Our study found that the majority of subjects, including those with mild and severe fibrosis, experienced regression of liver fibrosis. We also observed a significant decrease in the degree of hepatic fat accumulation.

The regression of liver fibrosis and steatosis remains a topic of debate, with some studies suggesting that the assessment method used plays a crucial role. Most studies use transient elastography to evaluate liver fibrosis [24]. However, in our study, we observed regression of fibrosis and steatosis in a substantial number of patients within a shorter time frame compared to other studies. This could be due to the use of Fibromax, which assesses liver function through biological parameters such as ALT, AST, and GGT. These values were found to change significantly three months after achieving SVR.

Despite our findings, it is important to note that our study has certain limitations. For instance, the sample size was small, and the monitoring period was short (9 months). Additionally, liver fibrosis and steatosis severity were assessed non-invasively only with Fibromax (no other non-invasive methods, such as transient elastography, were used), rather than through direct histological examination. Furthermore, we did not account for other factors that could have influenced liver fibrosis and steatosis. Another limitation is that the use of concurrent medications that may affect liver steatosis (such as statins, pioglitazone, or vitamin E) was not collected. Moreover, not all the elements of metabolic syndrome were available.

In our study, the percentage of subjects with chronic ethanol consumption was 11%, with a higher rate for males. Chronic alcohol consumption was considered more than 100 g of alcohol per week [25]. A study that included approximately 18,000 subjects treated with DAA, showed a percentage of 8.9% chronic ethanol consumers. However, the rates of achieving SVR were approximately equal in both subject groups, regardless of alcohol consumption, a finding also observed in our study, which supports clinical guidelines that do not recommend excluding alcohol-consuming patients from DAA treatment [26,27]. In another study, the prevalence of excessive alcohol consumption was 20% [28].

As observed in this study, the lowest BMI was found in patients with mild steatosis, and the highest in those with severe steatosis. These results are consistent with previous studies that report BMI and other anthropometric parameters factors associated with hepatic steatosis [29,30,31]. Moreover, regarding hepatic fibrosis, Ortiz et al. evaluated annual changes in the degree of hepatic fibrosis (rate of fibrosis progression) based on hepatic histology in HCV patients and highlighted that obesity is one of the main factors predicting disease progression [32].

In our study, similar findings were observed as in the literature, with the lowest BMI being more frequent in subjects with mild fibrosis and the highest in those with severe fibrosis. Therefore, in the current context of obesity, these findings demonstrate the importance of weight loss in the management of patients with HCV. After achieving a viral cure in patients with hepatitis C, weight loss can be beneficial for improving health status. Weight loss can improve metabolic health markers and reduce the risk of developing diabetes and other metabolic conditions. BMI may play an important role in the progression of hepatitis C and treatment success, and maintaining a healthy weight after a viral cure may be important for long-term health improvement [33,34].

The pathogenesis of hepatic steatosis in patients with HCV infection is not precisely established, although it is associated with both viral and metabolic factors. Obesity is associated with insulin resistance in peripheral glucose, called insulin resistance, and can lead to the development of T2DM and contribute to the development of hepatic steatosis and cardiovascular risk [35,36,37]. Diabetes and obesity are well-known risk factors in the evolution of liver disorders [38,39,40].

There is a clear association between triglyceride levels and chronic viral C hepatitis after treatment. In our study, a significant increase in triglyceride values was observed in both steatosis and hepatic fibrosis. Patients with hepatitis C may have elevated levels of triglycerides in the blood, and this may be associated with various metabolic conditions such as metabolic syndrome, nonalcoholic fatty liver disease, and diabetes mellitus. In patients successfully treated for hepatitis C, triglyceride levels may decrease significantly, which can be a sign of metabolic improvement. However, it is essential to continue monitoring triglyceride levels and taking measures to maintain them within a normal range as high levels can represent a risk factor for health problems.

Risk stratification models for HCC have been developed among SVR patients using pre-treatment data. However, the time from obtaining SVR represents a complexity that has not yet been taken into account. In some patients with liver cirrhosis, the resolution of hepatic fibrosis and hepatic remodeling after SVR may lead to a decline in the risk of HCC over time. However, some subjects, especially those with decompensated HCV-related cirrhosis, may not exhibit a resolution of fibrosis and a decrease in HCC risk after SVR. As time passes after SVR, the patient’s age increases and may acquire factors that decrease fibrosis or diminish the decrease in HCC risk after SVR (diabetes mellitus type 2, obesity, alcohol consumption), or even increase the risk of HCC among subjects without liver cirrhosis at the time of treatment.

Although the risk of HCC is significantly reduced after treatment with DAA, especially in patients with liver cirrhosis who achieve SVR, they still retain a high risk of HCC and thus require active surveillance. Biomarkers are necessary to identify those patients with the highest risk of HCC after a virologic cure. The optimal follow-up interval for patients with HCV after eradication treatment is not yet established.

## 5. Conclusions

After treatment for hepatitis C and achieving a viral cure, the risk of developing metabolic conditions may decrease. Our study showed a significant improvement in metabolic parameters in patients successfully treated with antiviral therapies. However, patients with a history of hepatitis C may still have an increased risk of developing diabetes and other metabolic conditions, even after a viral cure. Additionally, it is important to note that treatment for hepatitis C can be challenging for patients with pre-existing metabolic elements, as they may be more susceptible to adverse reactions and side effects of antiviral therapy.

## Figures and Tables

**Figure 1 metabolites-13-00571-f001:**
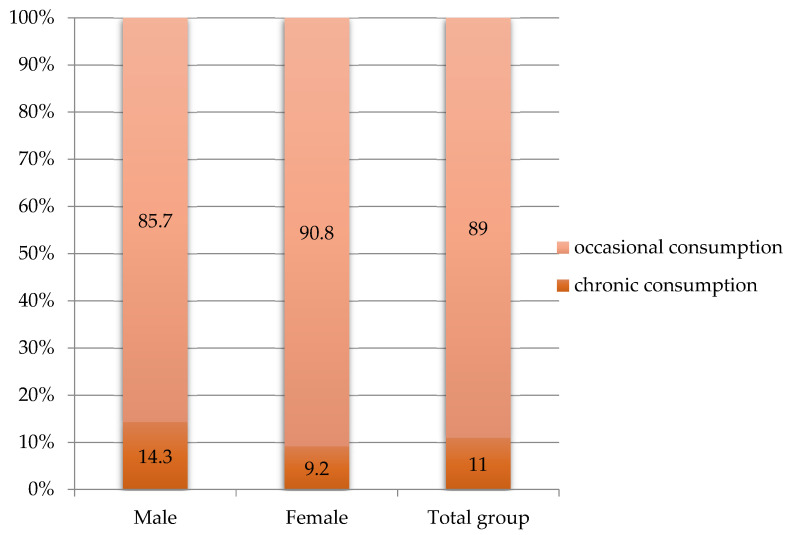
Alcohol consumption in the studied group.

**Table 1 metabolites-13-00571-t001:** Baseline characteristics.

Characteristics	Values
Sex	
Female	65.0%
Male	35.0%
Age	
Average ± SD	60.74 ± 8.58
Min-max/median/Skewness test values	35–77/61/−0.512
Fibrosis (FibroTest)	
Average ± SD	0.65 ± 0.18
Min-max/median/Skewness test values	35–77/61/−0.512
Degree of liver fibrosis	
F4	43.0%
F3	21.0%
F2	36.0%
Steatosis (SteatoTest)	
Average ± SD	0.50 ± 0.18
Min-max/median/Skewness test values	0.11–0.89/0.49/−0.015
Degree of liver steatosis	
S3	19.0%
S2	37.0%
S1	32.0%
S0	12.0%

F, fibrosis; S, steatosis.

**Table 2 metabolites-13-00571-t002:** Evolution of fibrosis and steatosis post-treatment as assessed by Fibromax.

KERRYPNX	T0	T3	Paired Samples Statistics
Fibrosis (FibroTest)- AUC	0.66 ± 0.18	0.55 ± 0.18	0.001
F2	0.45 ± 0.07	0.43 ± 0.13	0.205
F3	0.65 ± 0.04	0.51 ± 0.09	0.001
F4	0.82 ± 0.08	0.67 ± 0.16	0.001
Steatosis (SteatoTest)- AUC	0.50 ± 0.18	0.34 ± 0.14	0.001
S0	0.33 ± 0.14	0.30 ± 0.14	0.481
S1	0.34 ± 0.08	0.25 ± 0.11	0.001
S2	0.58 ± 0.06	0.40 ± 0.12	0.001
S3	0.76 ± 0.05	0.45 ± 0.11	0.001

T0, initial visit; T3, second visit; AUC, area under the curve.

**Table 3 metabolites-13-00571-t003:** Evolution of biological markers in patients with fibrosis.

Parameter	T0	T3	Difference from the Average Period	*p* for Paired Sample T Test
BMI (kg/m^2^)				
mild fibrosis severe fibrosis	25.96 ± 3.14 27.92 ± 5.14	27.18 ± 4.19 29.76 ± 4.66	+1.08 +1.64	0.410 0.183
*p* for F_ANOVA_ test	0.043	0.034		-
Fasting blood glucose (mg/dL)				
mild fibrosis severe fibrosis	106.58 ± 37.23 116.72 ± 48.66	102.81 ± 27.43 113.52 ± 38.28	−3.78 −3.20	0.331 0.583
*p* for F_ANOVA_ test	0.281	0.143		
Triglyceride (mg/dL)				
mild fibrosis severe fibrosis	105.28 ± 58.80 103.80 ± 35.74	126.17 ± 55.85 106.33 ± 39.23	+20.58 +2.53	0.013 0.617
*p* for F_ANOVA_ test	0.833	0.039		
GGT (U/L)				
mild fibrosis severe fibrosis	38.18 ± 22.83 87.22 ± 77.35	22.47 ± 9.50 34.28 ± 25.34	−15.71 −52.94	0.001 0.001
*p* for F_ANOVA_ test	0.001	0.010		
ALT (U/L)				
mild fibrosis severe fibrosis	74.24 ± 44.60 109.52 ± 75.46	26.64 ± 12.46 25.59 ± 15.05	−47.60 −83.93	0.001 0.001
*p* for F_ANOVA_ test	0.012	0.724		
AST (U/L)				
mild fibrosis severe fibrosis	51.81 ± 28.64 86.81 ± 50.69	22.28 ± 6.79 24.63 ± 8.32	−29.53 −62.18	0.001 0.001
*p* for F_ANOVA_ test	0.001	0.154		
LDL cholesterol (mg/dL)				
mild fibrosis severe fibrosis	123.83 ± 56.22 97.58 ± 32.49	125.67 ± 37.30 119.85 ± 38.66	+26.75 +16.30	0.175 0.504
*p* for F_ANOVA_ test	0.269	0.585		-

T0, initial visit; T3, second visit; GGT, gamma-glutamyl transferase; AST, aspartate aminotransferase; ALT, alanine aminotransferase; LDL cholesterol, low-density lipoprotein cholesterol; BMI, body mass index.

**Table 4 metabolites-13-00571-t004:** Evolution of biological markers in patients with steatosis.

Parameter	T0	T3	Difference from the Average Period	*p* for Paired Sample T Test
BMI (kg/m^2^)				
mild steatosissevere steatosis	25.39 ± 4.11 28.71 ± 4.49	29.62 ± 5.17 28.08 ± 4.05	+3.94 −0.84	0.004 0.457
*p* for F_ANOVA_ test	0.001	0.198		-
Fasting blood glucose (mg/dL)				
mild steatosissevere steatosis	102.91 ± 32.02 115.58 ± 20.45	100.59 ± 26.80 118.68 ± 25.11	−2.32 +3.11	0.497 0.584
*p* for F_ANOVA_ test	0.045	0.021		
Triglyceride (mg/dL)				
mild steatosissevere steatosis	88.89 ± 35.38 115.16 ± 44.45	106.00 ± 45.11 114.58 ± 46.11	+17.41 −0.58	0.001 0.950
*p* for F_ANOVA_ test	0.001	0.148		
GGT(U/L)				
mild steatosissevere steatosis	40.20 ± 23.77 105.21 ± 95.67	24.02 ± 15.77 31.95 ± 17.51	−16.18 −73.26	0.005 0.002
*p* for F_ANOVA_ test	0.001	0.011		
ALT(U/L)				
mild steatosissevere steatosis	58.64 ± 29.58 143.76 ± 69.84	23.27 ± 13.83 25.11 ± 9.22	−35.36 −118.66	0.001 0.001
*p* for F_ANOVA_ test	0.001	0.090		
AST(U/L)				
mild steatosissevere steatosis	47.64 ± 17.07 108.46 ± 50.09	23.67 ± 10.14 23.26 ± 6.05	−23.98 −85.20	0.001 0.001
*p* for F_ANOVA_ test	0.001	0.154		
LDL cholesterol(mg/dL)				
mild steatosissevere steatosis	98.75 ± 45.57 111.47 ± 44.89	119.48 ± 38.71 124.57 ± 37.65	+44.67 +5.48	0.064 0.692
*p* for F_ANOVA_ test	0.636	0.623		-

T0, initial visit; T3, second visit; GGT, gamma-glutamyl transferase; AST, aspartate aminotransferase; ALT, alanine aminotransferase; LDL cholesterol, low-density lipoproteins cholesterol; BMI, body mass index.

## Data Availability

Data are available from the corresponding author upon request. Data is not publicly available due to privacy.

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
