# Peer review of "The Influence of Metabolic Factors in Patients with Chronic Viral Hepatitis C Who Received Oral Antiviral Treatment"

_metabolites, 2023, doi:10.3390/metabo13040571_

Round 1

Reviewer 1 Report

Dear authors, 

your work is very well written and provide a lot of insights about an important topic.

I have just very few comments.

1. Did you have transient elastography? If not state it into the limitation section.

2. I feel that evolution of fatty liver and liver fibrosis in people who consumpe alcohol may be different from people who do not. Please provide more details and comments. 

3. Please add the references for the test you used to estimate liver disease.

4. Speculate about the possible changes between liver function and other organ function before and after DAA. 

5. Expand discussion about a possible tailored therapy for people with HCV and discuss about possible impact of mutation in genotype 1b on treatment success

6. Please underline the fact that surveillance post eradication should be perfomed for fatty liver but also for HCC.

Author Response

Reviewer: 1

Your work is very well written and provide a lot of insights about an important topic.

I have just very few comments.

Response 0: We would like to thank the Reviewer and the Editorial board for all the remarks regarding our work. We assure the EIC that we have read carefully every suggestion from this decision letter and tried our best to improve the quality of the document accordingly.

Point 1: Did you have transient elastography? If not state it into the limitation section.

 Response 1: Thank you for this important point. Indeed, we did not use transient elastography. The only method used in our study for evaluating fibrosis and steatosis was Fibromax (Fibrotest and Steatotest). We added this information in the limitation section.

Point 2: I feel that evolution of fatty liver and liver fibrosis in people who consume alcohol may be different from people who do not. Please provide more details and comments. 

Response 2: Thank you for this observation. Indeed, alcohol consumption enhances viral replication in liver diseases and accelerates disease progression. Even moderate amounts of alcohol can influence the degree of hepatic fibrosis and steatosis. Patients with liver disease are advised not to consume alcohol. We added this idea in the introduction section.

Point 3: Please add the references for the test you used to estimate liver disease.

Response 3: We thank the Reviewer for this very fine observation. We added the references for Fibromax test in the materials and methods section.

Point 4: Speculate about the possible changes between liver function and other organ function before and after DAA. 

 Response 4: Thank you for this great suggestion. Oral antiviral treatment for hepatitis may have some side effects on other organs, but these are usually minor. However, the treatment may be beneficial for other organs, as it can reduce inflammation and oxidative stress. Regarding side effects related to other organs, antiviral medications may have a negative effect on the kidneys, cardiovascular system, and central nervous system. However, these side effects are rare and usually only occur in patients who already have serious comorbidities. It is important that patients are closely monitored during antiviral treatment so that any side effects can be identified and treated. Overall, however, oral antiviral treatment is considered safe and effective in treating hepatitis and has significant benefits for  overall health. We added this idea in the introduction section.

Point 5: Expand discussion about a possible tailored therapy for people with HCV and discuss about possible impact of mutation in genotype 1b on treatment success.

 Response 5: We thank the Reviewer for this observation. The oral antiviral treatment for patients with chronic viral C hepatitis is adapted according to current guidelines for this pathology (treatment for 8, 12, or 24 weeks depending on the severity of the disease). There are six genotypes and over 100 subtypes of HCV known. Among these, genotypes 1, 2, and 3 are found worldwide, and in Romania, genotype 1b is the most commonly encountered, in a percentage of 99%. In our research, all the subjects presented genotype 1b. These data are supported by other epidemiological studies that highlight the higher prevalence of genotype 1, followed by genotype 3 (Nahon et al., 2017). Studies show that a much higher prevalence of hepatic steatosis is encountered in HCV genotype 3 infection, compared to non-genotype 3 HCV (74% versus 48%) (Lonardo et al., 2006). We added this information in the introduction and discussion section.

Point 6: Please underline the fact that surveillance post eradication should be perfomed for fatty liver but also for HCC.

 Response 6: Thank you for pointing this extremely important aspect out. Although the risk of HCC is significantly reduced after treatment with DAA, especially in patients with liver cirrhosis who achieve SVR, they still retain a high risk of HCC and thus require active surveillance. Biomarkers are necessary to identify those patients with the highest risk of HCC after virologic cure. The optimal follow-up interval for patients with HCV after eradication treatment is not yet established. We added this idea in the discussion section.

Reviewer 2 Report

This is a study that assessed changes in metabolic parameters and liver steatosis and fibrosis (as evaluated by the blood test Fibromax) before and after treatment among patients with hepatitis C virus. My comments/questions are as follows:

1. In the Methods section, please provide citations regarding the sensitivity and specificity of Fibromax in assessing steatosis and fibrosis. Please cite studies to support the validity and reliability of this test, as it is the key outcome measure of the paper. Please also provide citation for lines 110-122 in the Methods.

2. Only patients with genotype 1b were included. Please provide a brief explanation why patients with other genotypes were not included in the study.

3. Table 1: Please clarify 'Average Fibrosis' -- was this the average FibroTest score? 

4. Table 2: Suggest to clarify in the title 'Evolution of fibrosis and steatosis post-treatment as assessed by Fibromax'. Similarly, please indicate the units and tests used within the table for Fibrosis and Steatosis. 

5. Figure 1: Please indicate in the Discussion the definition of 'chronic alcohol consumption.' Furthermore, is there data on the amount of alcohol used by the patients?

6. Table 3 included various lab tests such as triglycerides, AST, ALT, etc. Please indicate in the Methods whether these were performed with patients on a fasting state. Also indicate briefly whether these were send-out tests to a certified laboratory. Please clarify 'glycemia' (did the authors mean 'fasting blood glucose'? 

7.  For all of the parameters in Tables 3 and 4, please indicate the unit of measurement (mg/dL, etc).

8. Please provide citations for Lines 267 to 275

9. Do we have data on the concurrent use of medications, such as statins, vitamin E, and pioglitazone? The use of these medications may impact metabolic parameters and must be mentioned in the discussion. If these are not available, they must be stated as a limitation of the study.

10. In the conclusion, the authors used 'metabolic syndrome' multiple times. However, the definition of 'metabolic syndrome' includes waist circumference, blood pressure, and HDL cholesterol levels -- all of which were not mentioned in the paper. The conclusion must be revised to state 'metabolic parameters' as opposed to 'metabolic syndrome.' Furthermore, if waist circumference, blood pressure, and HDL cholesterol levels are available, these should be included in the tables. If these are not available, it must be stated as a Limitation.   

Author Response

Reviewer: 2

This is a study that assessed changes in metabolic parameters and liver steatosis and fibrosis (as evaluated by the blood test Fibromax) before and after treatment among patients with hepatitis C virus. My comments/questions are as follows:

Response 0: We would like to thank the Reviewer and the Editorial board for all the remarks regarding our work. We assure the EIC that we have read carefully every suggestion from this decision letter and tried our best to improve the quality of the document accordingly.

Point 1: In the Methods section, please provide citations regarding the sensitivity and specificity of Fibromax in assessing steatosis and fibrosis. Please cite studies to support the validity and reliability of this test, as it is the key outcome measure of the paper. Please also provide citation for lines 110-122 in the Methods.

Response 1: Thank you for this very good remark. We added citations regarding Fibromax in the materials and methods section.

Point 2: Only patients with genotype 1b were included. Please provide a brief explanation why patients with other genotypes were not included in the study.

 Response 2: Thank you for this pertinent observation.  There are six genotypes and over 100 subtypes of HCV known. Among these, genotypes 1, 2, and 3 are found worldwide, and in Romania, genotype 1b is the most commonly encountered, in a percentage of 99%. In our research, all the subjects presented genotype 1b. These data are supported by other epidemiological studies that highlight the higher prevalence of genotype 1, followed by genotype 3 (Nahon et al., 2017). Studies show that a much higher prevalence of hepatic steatosis is encountered in HCV genotype 3 infection, compared to non-genotype 3 HCV (74% versus 48%) (Lonardo et al., 2006). We added this information in the introduction and discussion section.

Point 3: Table 1: Please clarify 'Average Fibrosis' -- was this the average FibroTest score? 

Response 3: Thank you for this remark. We added in Table 1 the name of the tests: FibroTest and  SteatoTest.

Point 4: Table 2: Suggest to clarify in the title 'Evolution of fibrosis and steatosis post-treatment as assessed by Fibromax'. Similarly, please indicate the units and tests used within the table for Fibrosis and Steatosis. 

 Response 4: Thank you for this very important observation. We changed the title for Table 2 and we also mentioned the tests used (FibroTest and SteatoTest) and the units.

Point 5: Figure 1: Please indicate in the Discussion the definition of 'chronic alcohol consumption.' Furthermore, is there data on the amount of alcohol used by the patients?

 Response 5: Thank you for this point. We added in the discussion and methods section data regarding the amount of alcohol used by the subjects.

Point 6: Table 3 included various lab tests such as triglycerides, AST, ALT, etc. Please indicate in the Methods whether these were performed with patients on a fasting state. Also indicate briefly whether these were send-out tests to a certified laboratory. Please clarify 'glycemia' (did the authors mean 'fasting blood glucose'? 

 Response 6: We thank the Reviewer for this observation. We added these information in the methods section. The usual blood tests were performed in the hospital with patients on a fasting state. Fibromax test was performed within the BioPredictive laboratories. We changed glycemia in “fasting blood glucose”.

Point 7:  For all of the parameters in Tables 3 and 4, please indicate the unit of measurement (mg/dL, etc).

 Response 7: Thank you for this suggestion. We added the unit of measurement for Tables 3 and 4.

 Point 8: Please provide citations for Lines 267 to 275

 Response 8: Thank you for this observation. We added citations for the mentioned lines.

Point 9: Do we have data on the concurrent use of medications, such as statins, vitamin E, and pioglitazone? The use of these medications may impact metabolic parameters and must be mentioned in the discussion. If these are not available, they must be stated as a limitation of the study.

Response 9: Thank you for pointing this important aspect out. These data are not available in our study. We added this aspect in the study limitation section.

 Point 10: In the conclusion, the authors used 'metabolic syndrome' multiple times. However, the definition of 'metabolic syndrome' includes waist circumference, blood pressure, and HDL cholesterol levels -- all of which were not mentioned in the paper. The conclusion must be revised to state 'metabolic parameters' as opposed to 'metabolic syndrome.' Furthermore, if waist circumference, blood pressure, and HDL cholesterol levels are available, these should be included in the tables. If these are not available, it must be stated as a Limitation.   

 Response 10: Thank you for this useful remark. Now all the elements of metabolic syndrome are available and we added this idea in the limitation section. We changed 'metabolic syndrome' to 'metabolic parameters'.

Round 2

Reviewer 2 Report

The authors have adequately addressed my inquiries/comments. There are just some typographical errors/minor corrections:

-- Line 120-121: "a jeun" seems like a typographical error

-- Line 289: "use of medication was not included..." must be revised to "use of concurrent medications that may affect liver steatosis (such as statins, pioglitazone, or vitamin E), were not collected..."

-- Line 351: "metabolic parameters markers" ('markers' should be deleted)

-- There are a few other minor spelling errors throughout the document

Once these minor issues have been addressed, I believe this paper may proceed to publication.
